# High-Throughput ^1^H-Nuclear Magnetic Resonance-Based Screening for the Identification and Quantification of Heartwood Diterpenic Acids in Four Black Pine (*Pinus nigra* Arn.) Marginal Provenances in Greece

**DOI:** 10.3390/molecules24193603

**Published:** 2019-10-07

**Authors:** Kostas Ioannidis, Eleni Melliou, Prokopios Magiatis

**Affiliations:** 1Laboratory of Forest Genetics and Biotechnology, Institute of Mediterranean and Forest Ecosystems, Hellenic Agricultural Organization “Demeter”, Ilissia, 11528 Athens, Greece; 2Department of Pharmacognosy and Natural Products Chemistry, Faculty of Pharmacy, University of Athens, Panepistimiopolis Zografou, 15771 Athens, Greece; emelliou@pharm.uoa.gr (E.M.); magiatis@pharm.uoa.gr (P.M.)

**Keywords:** quantitative nuclear magnetic resonance, high-throughput screening, resin acids, diterpenes, *Pinus nigra*, provenances

## Abstract

A high-throughput quantitative Nuclear Magnetic Resonance ^1^H-NMR method was developed and applied to screen the quantity of the diterpenic resin acids in the heartwood of black pine, due to the renewed scientific interest in their medicinal properties and use in various diseases treatment. The 260 samples were taken from *Pinus nigra* clones, selected from four provenances of the Peloponnese (Greece), participating in a 35-year-old clonal seed orchard. Total resin acids per dry heartwood weight (dhw) varied greatly, ranging from 30.05 to 424.70 mg/g_dhw_ (average 219.98 mg/g_dhw_). Abietic was the predominant acid (76.77 mg/g_dhw_), followed by palustric acid (47.94 mg/g_dhw_), neoabietic acid (39.34 mg/g_dhw_), and pimaric acid (22.54 mg/g_dhw_). Dehydroabietic acid was at moderate levels (11.69 mg/g_dhw_), while levopimaric, isopimaric, and sandaracopimaric acids were in lower concentrations. The resin acid fraction accounted for 72.33% of the total acetone extractives. Stilbenes were presented in significant quantities (19.70%). The resin acid content was composed mainly of the abietane type resin acids (83.56%). Peloponnesian *Pinus nigra* heartwood was found to be the richest source of resin acids identified to date and is considered the best natural source for the production of such bioactive extracts. The results indicate a high potential for effective selection and advanced breeding of pharmaceutical and high economic value bioactive substances from *Pinus nigra* clones.

## 1. Introduction

In Greece, as well as in other European countries, black pine constitutes extensive natural forests. Due to its advantages, it is considered one of the most important silvicultural coniferous species and it is extensively used in reforestation programs throughout the country [1]. Its significance lead to its planting outside its natural range, e.g., North and South America, Australia, and New Zealand. Furthermore, its importance is verified by its potential use for the production of high added value products, i.e., bioactive compounds produced from wood and wood waste materials [2,3]. Such bioactive substances are, inter alia, the resin acids occurring in pine’s oleoresin.

Oleoresin consists of a non-volatile fraction, also referred as colophony or rosin, dissolved in a volatile monoterpenic part, the turpentine. Oleoresin, in coniferous species, is synthesized by the epithelial cells surrounding the resin canals, from which it is exuded as a response to mechanical wounds or biotic attacks, thus constituting a defense mechanism of living trees [4]. Resin acids, among other substances, are part of the non-volatile fraction, composed of a mixture of diterpenic resinous compounds. Diterpenic resin acids are the most common members of oleoresin [4] and the most abundant compounds in heartwood and knots [5,6,7,8]. Their fractions can vary among individual trees and species [9,10,11]. Many softwoods, and particularly *Pinus* species, contain relatively high oleoresin amounts, moving through the wood’s extensive vertical and radial networks of resin canals. 

Resin acids are mostly tricyclic compounds, arranged in two types depending on the presence or absence of double bonds in their aromatic rings. The first, called abietane-type, includes acids such as abietic acid (**1**), dehydroabietic acid (**2**), neoabietic acid (**3**), palustric acid (**4**) and levopimaric acid (**5**), and the second, named pimarane-type, includes acids such as pimaric acid (**6**), sandaracopimaric acid (**7**), and isopimaric acid (**8**) (Figure 1). The cyclic diterpene acids originate from a common acyclic biosynthetic precursor, geranylgeranyl diphosphate (GGPP) [12]. Diterpene synthase enzymes act on the 20-carbon GGPP substrate to form diterpenes, which are subsequently hydroxylated and then oxidized, in two independent transformations, by other enzymes to the diterpene acids [13]. 

The diterpene synthase genes involved in resin acid formation prove their highly genetically-correlated heartwood properties [14,15]. Selection and breeding in *Pinus* species could result in doubling the oleoresin yields [16] and further genetic gains in extractive yield could be achieved [17]. Genetic control of extractive production has considerable economic potential in some species [18].

Pine resin has been highly applied in industry. Resin acids and their derivatives, apart from being used in intermediate chemical products, such as polymer additives, tackifiers, emulsifiers in synthetic rubber, paper sizing to control water absorptivity, and aroma industry, are also deployed in adhesives, surface coatings, printing inks, chewing gums, and in the Greek wine industry [19,20]. Furthermore, several medicinal uses concerning the treatment of abscesses, boils, cancers, toothache, and skin diseases, such as psoriasis and ringworm [19], have been reported even from antiquity. According to Dioscorides, during the first century A.D., a wine called retsina, i.e., wine that had been mixed with resin from *Pinus halepensis*, was used for lung and stomach disorders and as an antidote for cough [19].

In recent years, many studies have shown that several substances in rosin have antimicrobial properties [21], cardiovascular effects [22,23], antiallergic properties [24], anti-inflammatory properties [25,26], anticonvulsant effects [27], antiulcer-gastroprotective properties [28,29], and cytotoxic activity toward human fibroblasts [29]. Their medicinal properties, due to the bioactivity of natural abietane and pimarane diterpene acids, which have been reviewed by San Feliciano et al. [30] and Reveglia et al. [31] respectively, have caused a renewed scientific interest in pine extracts.

The aim of the present work was to analyze the resin acids composition and variability from the heartwood extractives of four *Pinus nigra* L. subsp. *pallasiana* marginal provenances from the Peloponnese, southern Greece, growing in a clonal seed orchard. The heartwood of these trees produced an exceptionally high extractive content, approximately 30% *w*/*w* [3], making it particularly interesting for the present analysis. The target compounds were the abietane-type resin acids: abietic, dehydroabietic, neoabietic, palustric, and levopimaric acids, and the pimarane-type: pimaric, isopimaric, and sandaracopimaric acids. The ^1^H-NMR spectroscopy was used to analyze, qualitatively and quantitatively, the highly complex diterpenic acid mixtures of heartwood rosin.

Many studies have been carried out on pine oleoresin composition, some of which have demonstrated the potential of NMR spectroscopy, a modern analytical methodology with applications in complex mixtures [32,33]. The applied technique highlights its feasibility and wide practical applicability in characterizing various complex matrixes, from leaves [34] and walnuts [35] for profiling “green” extracts and different genotypes, respectively, to foodstuffs [36,37] for quality/safety or discrimination purposes, and biological [38] for lipid profiling. ^13^C-NMR spectroscopy has already been used for the quantitation of resin acids in wood extracts from *Pinus radiata* [39] and *Pinus nigra* ssp. *laricio* [20,40]. Skakovskii et al. [41], through the use of high-field NMR spectrometers introduced the alternative ^1^H-NMR analysis of resin acids, and this technique is here preferred as quicker and straightforward. Moreover, the possible simultaneous record of ^1^H-NMR and ^13^C-NMR experiments can pave the way to the robust analysis of resin acid by combined data as performed for other systems [42].

## 2. Results

^1^H-NMR spectrum of black pine (P. nigra Arn.) heartwood extracts, showing the characteristic peaks of the studied resin acids and internal standard, are presented in Figure 2. The whole spectrum is available as Appendix A. Their chemical shifts are presented in Table 1. Recovery and intraday precision data are presented in Table 2 and Table 3, respectively.

In the frame of this investigation 260 samples were analyzed. The heartwood of the studied trees was exceptionally rich in acetone extractive content (TAE), averaging 304.15 mg/g of dry heartwood (dhw), which contained high resin acid quantities. The mean resin acid fraction accounted for 72.33% of the total extractive content, while stilbenes were present in significant quantities, comprising 19.70%. The remaining 7.97% is referred to as other substances present in the TAE, i.e., minor unidentified resin acids, fatty acids, unsaponifiables, triglycerides, other phenols, waxes, sterols, etc.

The mean concentration of all studied diterpenic resin acids as their min and max values are presented in Table 4. The total resin acid (TRA) content showed an extent variation ranging from 30.05 to 424.70 mg/g_dhw_, with an average of 219.98 mg/g_dhw_ (±96.20). Abietic acid was the predominant acid (76.77 ± 37.39 mg/g_dhw_), followed by palustric acid (47.94 ± 23.31 mg/g_dhw_) and neoabietic acid (39.34 ± 21.21 mg/g_dhw_), all belonging to the abietane-type resin acids. The next in abundance acid was the pimaric acid, with a mean concentration of 22.54 ± 11.28 mg/g_dhw_. Dehydroabietic acid was found at moderate levels (11.69 ± 5.73 mg/g_dhw_), while the rest, levopimaric acid, isopimaric acid, and sandaracopimaric acid, were observed in lower concentrations, i.e., 8.07 ± 211.38 mg/g_dhw_, 10.91 ± 6.53 mg/g_dhw_, and 2.72 ± 1.49 mg/g_dhw_, respectively.

In Table 5, the mean resin acid concentrations, the results from the analysis of variance (ANOVA), as well as Duncan’s Multiple Range Test (MRT) results, of the four *Pinus nigra* L. provenances are presented. The ANOVA and MRT showed that there were statistically significant (*p* < 0.001) differences among clones for all examined acids (data not shown) and also among provenances in concentrations of both abietic acid (*p* < 0.01) and pimaric acid (*p* < 0.01).

In Figure 3, the mean percentage of each individual resin acid to total resins, as well as their types, are presented. Abietic, palustric, and neoabietic acids, which were the most abundant resin acids, accounted for 34.90%, 21.79%, and 17.88%, respectively. Dehydroabietic and levopimaric acids comprised 5.32% and 3.67% of the TRA, respectively. Pimaric acid was in a fairly large quantity, corresponding to 10.25% of the total studied resin acids, while sandaracopimaric and isopimaric acids corresponded to 1.24% and 4.96%, respectively, of the total studied resin acids. The resin acid content was composed mainly of the abietane resin acid type (≈84%), while the pimarane-type accounted only for the rest, ≈16%, so the abietane- to pimarane-type resin acids ratio was circa 5.2. In Figure 4, the mean percentage content of the constituents of black pine’s heartwood acetone extraction from the Peloponnese is presented.

In Figure 5, the frequency distribution of the total resin acids (TRA) of black pine from the Peloponnese are presented.

## 3. Discussion

The current study aimed to demonstrate that ^1^H-NMR spectroscopy can be effectively used for the analysis of *Pinus nigra* heartwood resin acids although their spectra are complicated and partially overlapping. Despite the structural similarity of the terpenic acids included in the oleoresin, we were able to select non overlapping peaks, belonging to specific protons, that could lead to the identification and quantitation of the target resin acids. Using ^1^H-NMR spectroscopy, we achieved to minimize time-consuming procedures of separation of every analyte before quantitation.

As it was previously shown in the cases of olive oil [43,44], beer [45], wine [33], and stilbenes from *Pinus nigra* heartwood [3], qNMR is the most appropriate method for quantitation. The developed methodology for the analysis of heartwood resin acids requires less than a minute to record the spectrum and perform the analytes quantitation, whereas using the previously reported ^13^C-NMR methodology requires a considerably longer time, i.e., approximately two hours [40] or even more [46]. The advantage of the current methodology allowed the analysis of a very large amount of samples in a short period and led to reliable quantitative data.

The results indicated obvious individual tree and clone variation in heartwood resin acids content due to genetic and environmental factors affecting rosin production and secretion [4,47]. Venäläinen et al. [48] also reported high variation among trees in the resin acid concentrations of *Pinus sylvestris*. Heartwood resin acid contents present provenance variation as well (Table 5) because of the heterogeneity among populations [3]. In general, the southeastern provenance of Parnonas, grown at the most xerothermic environment [49] and subjected to the strongest winds [50], showed higher values of total resin acids, i.e., for neoabietic acid, palustric acid, and sandaracopimaric acid, while the southwestern of Taigetos had the lowest concentrations of all except for dehydroabietic and levopimaric acids. For the latter, Taigetos provenance showed the highest content of all provenances. Although Feneos provenance (northeastern origin) predominated in dehydroabietic acid and pimaric acid concentrations, only the latter acid statistically differed from Taigetos origin. The origin of Zarouhla presented the highest value for abietic and isopimaric acids, having a statistically significant difference only for the abietic acid found in Taigetos. Willför et al. [51] found similar results, mentioning that there were no significant differences between trees from the southern and northern regions, thus confirming Song’s [52] view that the oleoresin of trees within several *Pinus* species show similar basic chemical characteristics, regardless of their geographic location. 

The present study stated that acetone extractives of *Pinus nigra* heartwood mainly consisted of resin acids. These results were in accordance with those of Martínez-Inigo et al. [53], Willför et al. [51], and Belt et al. [54] for *Pinus sylvestris*. Hafizoğlu [55], in a review article, mentioned that several researchers have found that most of the oleoresin in *Pinus nigra* and *P. sylvestris* were comprised of resin acids. In the same direction were the results of Uprichard and Lloyd [56] in *P. radiata*, reporting that resin acids predominate in heartwood total extractives. Yildirim and Holmbom [57] have found a remotely lower percentage in stemwood extractives of *Pinus brutia*.

Lower resin acid concentrations compared to the present study were detected in *Pinus sylvestris* by Martínez-Inigo et al. [53], Hovelstad et al. [58], and Belt et al. [54]. Concerning *Pinus sylvestris* heartwood, Harju et al. [59], Venäläinen et al. [48], Willför et al. [51], and Arshadi et al. [60] estimated lower resin acid content, too. Moreover, in *Pinus nigra*, Yildirim and Holmbom [61] found similar results with the previous researchers. Correspondingly, medium resin acid concentrations were found in *P. radiata* by Lloyd [7]. On the contrary, Willför et al. [62] found in both *Pinus nigra* and *P. brutia* heartwood even lower amounts of resin acids than the present study, without referring to predominant acids or the fatty/resin and abietane/pimarane ratios. Low levels of resin acid were noted in *Pinus contorta* by Lewinsohn et al. [63] and Arshadi et al. [60]. Benouadah et al. [64] surprisingly found even lower concentrations in dry *Pinus halepensis* heartwood. 

The fatty acid/resin acid ratio in the present study was estimated as very low due to the high presence of resin acids, as well as stilbenes, in total acetone extractives and could not exceed 0.18. Lower fatty/resin acid ratio in *Pinus sylvestris* was reported by Martínez-Inigo et al. [53] and Belt et al. [54], depending on the heartwood sample position, and in *P. radiata* by Uprichard and Lloyd [56]. Arshadi et al. [60] reported that in *Pinus sylvestris* and *Pinus contorta*, in general, the concentration of resin acids was higher than that of fatty acids. In *Pinus contorta* and *Pinus attenuata*, the results of Anderson et al. [65] confirmed the above findings as well as those of the present study. In contrast, Hafizoğlu’s [55] review article reported that fatty/resin acid ratio was a multiple of the unit concerning *P. nigra* and *P. sylvestris*, and that only in *P. brutia* the resin acids were double the fatty ones. The results of Uçar and Fengel [66], as well as those of Uçar and Balaban [67], for *Pinus nigra* were in agreement with Hafizoğlu’s. In both studies, the fatty/resin ratio depended on the *Pinus nigra* variety or provenance. 

In the present study, the abietane-type dominated over the pimarane-type resin acids and the estimated (abietane/pimarane) ratio was calculated at 5.1. This is consistent with the results of Harju et al. [68] and Willför et al. [51] in *Pinus sylvestris*, who estimated a little higher ratio values, as well as with Uçar and Fengel [66], who estimated a little lower ratio values, depending on variety. The common element in all studies was that the resin acids of abietane-type were multiples compared to the pimarane ones.

Abietic acid was the most abundant resin acid, which is in accordance with the findings of Martínez-Inigo et al. [53], Willför et al. [51,62], and Yildirim and Holmbom [61] in *Pinus sylvestris*, of Yildirim and Holmbom [57] in *Pinus brutia*, and of Anderson et al. [65] in *Pinus attenuata*. Hafizoğlu’s review article [55] reported that, in most cases, abietic, mainly in *Pinus nigra*, and levopimaric, mainly in *Pinus sylvestris*, acids were the most abundant, followed by palustric acid though dependent on species and origins. Abietic acid was found to be the most abundant resin acid in Scots pine according to Harju et al. [68], but followed by the mixture of palustric/levopimaric acids and neoabietic or pimaric acids. Abietic acid was the most abundant resin acid in *P. radiata* too [56]. Different results were those of Rezzi et al. [20] who distinguished two clusters of *Pinus nigra* ssp. *laricio*. In the first one, levopimaric acid was the most abundant resin, whereas in the second, dehydroabietic acid and levopimaric acid were approximately in equal concentrations. Cannac et al. [46] identified *Pinus nigra* ssp. *laricio*, from Corsica too, as belonging to the first cluster.

In the examined *Pinus nigra* provenances, abietic acid was the most abundant one, followed by palustric acid, neoabietic acid, and pimaric acid. Similar results, but with a few differences in the classification of resin acid concentrations, were also those of Ekeberg et al. [69] in Scots pine and Benouadah et al. [64] in Aleppo pine. In both studies, abietic acid was the major acid though dehydroabietic acid was ranked second or third. Papajannopoulos et al. [70] reported that the basic composition of the oleoresin of three Greek pine species, *Pinus halepensis*, *P. brutia*, and *P. pinea*, were found to be similar concerning the abundance of several resin acids. The mixture of palustric/levopimaric acids were the most abundant followed by the abietic, in similar concentrations, and neoabietic or isopimaric acids. The above were in accordance with the results of Lange and Stefanovic Janezic [71] for *Pinus sylvestris* and Lloyd [7] for *Pinus radiata*.

Different were the results of Uçar and Fengel [66] who found palustric acid to be the richest resin acid in the stemwood of *Pinus nigra* varieties, followed by isopimaric acid or dehydroabietic acid (depending on the variety), while abietic acid was ranked third. Palustric acid was found to be the most abundant *Pinus nigra* resin acid by Uçar and Balaban [67] followed by abietic acid, neoabietic acid, or levopimaric acid, depending on the individual tree.

The results of Lewinsohn et al. [63] obtained in the case of *Pinus contorta* are in contrast with our results. They found that the total resin acids consisted mainly of levopimaric acid, followed by palustric, isopimaric, abietic, and neoabietic acids at moderate levels. The above composition resembled those of mature *Pinus contorta* rosin stated by Anderson et al. [65]. In *Pinus attenuata*, Anderson et al. [65] found that dehydroabietic acid was the major resin acid followed by the mixture of levopimaric/palustric acids, and finally by neoabietic acid.

The present study sets a framework for future research on a number of issues, which could lead to new scientific results. Exploiting the individual clone and provenance variation of the genetic material of *Pinus nigra* from the Peloponnese, its heartwood has been found to be the richest source in resin acids, as well as in stilbenes [3], identified to date. Effective selection and advanced breeding require further research towards the estimation of several genetic parameters (i.e., heritability, breeding values, and genetic gains for all the above mentioned traits) of the studied provenances and clones.

## 4. Materials and Methods 

### 4.1. Plant Material

The plant material was sampled from a 10 hectares *Pinus nigra* Arnold clonal seed orchard (CSO), established in the western part of the Peloponnese in 1978. The CSO comprises 52 clones and a total number of 2700 grafts, derived from intensively selected plus trees, originating from four marginal provenances (Zarouhla, Feneos, Parnonas, and Taigetos) of the natural black pine forest of the Peloponnese (Figure 6). Clones (one ramet/clone) were randomly assigned at 6 m × 6 m spacing within replications (single tree plot design), without blocking, with the only restriction that no grafts of the same clone were planted closer than 30 m [72]. 

### 4.2. Sampling

Sampling, coring, heartwood discrimination and orientation, and extraction protocol are extensively described in Ioannidis et al. [3]. In brief, increment cores, approximately 30 cm above ground and in a north-south orientation, of 12 mm in diameter were extracted from a total of 260 healthy individuals that were sampled during October and November 2013 and covered all 52 clones participating in the seed orchard (five ramets per clone), and stored in darkness at −76 °C. Heartwood was separated from the rest of the core, using the benzidine discrimination method, and milled to produce ≤0.75 mm particles, which were freeze-dried for 48 h at −52 °C and 0.03 mbar pressure, to ensure almost complete removal of moisture and volatile compounds. 

### 4.3. Extraction Protocol 

In brief, resin acids were extracted with 12 ml acetone from 200 mg (±0.1 mg) of freeze-dried, ground heartwood. The mixture was first placed in an orbital shaker at 350 rpm (Edmund Bühler GmbH, Bodelshausen, Germany) in darkness at room temperature for 24 h, followed by its transfer to an ultrasonic bath (Semat, UK) for 1 h to complete the extraction. The liquid phase was separated by centrifuging (3075× *g* for 15 min) (Eppendorf 5810R, Germany) and the solvent was evaporated in a heated vacuum rotary evaporator (Buchi, Switzerland) at 40 °C to determine the weights. Recovery of the first extraction was calculated by employing successive extraction procedures and quantitation in each one.

#### 4.3.1. H-NMR Spectral Analysis/Quantitation Methodology 

The dried extractives were submitted to chemical analysis by ^1^H-NMR using syringaldehyde as internal standard and deuterated chloroform (CDCl_3_) as solvent. The extract of each sample obtained from the extraction procedure was dissolved in 600 μL deuterated chloroform (CDCl_3_) (Euriso-Top) and the solution was transferred to a 5 mm NMR tube. ^1^H-NMR spectra were recorded at 400 MHz (Bruker DRX400). Typically, 16 scans were collected into 32K data points over a spectral width of 0–16 ppm with a relaxation delay of 10 s. Prior to Fourier transformation (FT), an exponential weighting factor corresponding to a line broadening of 0.3 Hz was applied. The spectra were phase corrected automatically using TopSpin software (Bruker, Billerica, MA, USA). For the peaks of interest, accurate integration was performed manually. Quantitative determination of the resin acids (RA), i.e., the concentration C_RA_, was obtained by comparing the area of the selected signals with that of the internal standard (IS), i.e., the relative integration (I_RA_). The following formula was used, which include I_RA_ as well as the weight (m_sample_ = 200 mg), the molecular weight (MW_RA_), and the average recovery of each resin acid (R_RA_, Table 2) of the analyzed dry heartwood powder: C_RA_ = [(0.002745 × I_RA_ × MW_RA_)/m_sample_]/R_RA_

The recovery was calculated as the ratio of successive extractions of heartwood samples for the studied resin acids. The samples contained different levels of extractives in order to obtain unbiased results. The intraday precision (i.e., the experimental error) was determined by analyzing three replicates of three random samples in the same day. Concentrations were based on freeze-dried heartwood (dhw) and expressed in mg/g_dhw_. Syringaldehyde (Acros Organics) internal standard (IS) solution was prepared in acetonitrile at a concentration of 0.5 mg/mL. Prior to use, the IS solution reached room temperature and the solutions of all the extracted samples were mixed, before evaporation, with 1 mL of a syringaldehyde solution, i.e., 0.00274 moles of syringaldehyde.

The identity of all compounds was defined by the literature data [41,73,74,75]. Due to rather complicated resin acids’ ^1^H-NMR spectra, nonoverlapping, undoubtedly defined peaks of the olefinic or aromatic protons were selected for quantitation. Concerning the abietane-type resin acids, abietic acid (1), neoabietic acid (2), dehydroabietic acid (3), palustric acid (4), and levopimaric acid (5) were distinguished by their single C(H)-14 proton at 5.77 ppm, 6.20 ppm, 6.88 ppm, 5.39 ppm, and 5.53 ppm, respectively. For the pimarane-type resin acids, pimaric acid (6) and isopimaric acid (8) were distinguished by the C(H)-15 proton at 5.67 ppm and 5.84 ppm, respectively, and sandaracopimaric acid (7) was distinguished by the C(H)-14 proton at 5.22 ppm. It should be noted that for pimaric and isopimaric acids the selected signal for quantitation was only one of the four equivalent peaks of the corresponding doublet to avoid overlapping with other peaks. In these two cases, the integration value was multiplied by a factor of four before comparing the area with that of the internal standard in order to estimate the concentration.

To exclude the possibility of overlapping peaks we have performed 2D HMQC (2 Dimensional Heteronuclear Multiple Quantum Correlation) experiment presented in Appendix A.

#### 4.3.2. Statistical Analysis 

Descriptive statistics, analysis of variance (ANOVA), as well as Duncan’s Multiple Range Test (MRT), based on the 0.05 level of significance, were performed in order to check the hypothesis that there were statistically significant differences among the mean concentrations in the samples of all clones and among the four provenances, using SPSS v.20 software for Windows (IBM SPSS Statistics 2011, IBM Corp. New York, NY, USA).

## 5. Conclusions

The ^1^H-NMR spectroscopy proved to be a fast, sensitive, accurate, and comfortable methodology for high-throughput identification and quantitation of resin acids. According to this study, the genetic material of black pine heartwood, originating from the Peloponnese, was proved to be the richest natural source of resin acids compared to any other European pine species/population referred to in the literature. The high resin acid concentrations, as well as differences among origins for some of them, can be attributed to their adaptation over time to individual environments that differ in bioclimatic and soil parameters. Such marginal populations growing in the edge of their natural range, under the pressure and impact of climate change, will be populations for further investigations and adaptability testing. Furthermore, the presence of great variation in resin acid concentrations among trees and origins from the Peloponnese, as well as the consensus of genetic control of resin acid production-segregation, highlight the great potential for effective selection and advanced breeding of the studied genetic material for pharmaceutical and high economic value bioactive substances derived from *Pinus nigra* L. heartwood. 

## Figures and Tables

**Figure 1 molecules-24-03603-f001:**
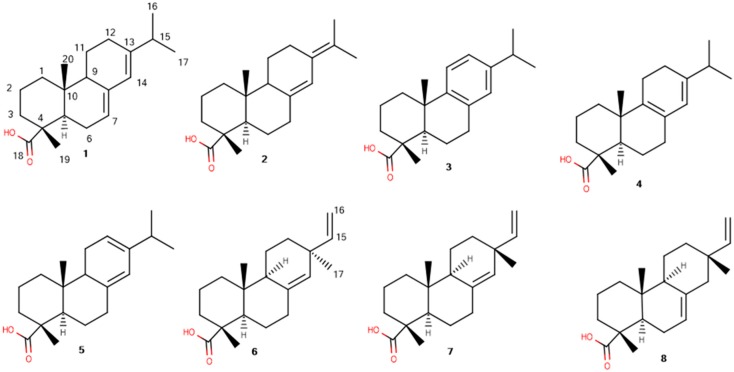
The structures and numbering of the diterpenic resin acids of the *Pinus nigra* heartwood extracts.

**Figure 2 molecules-24-03603-f002:**
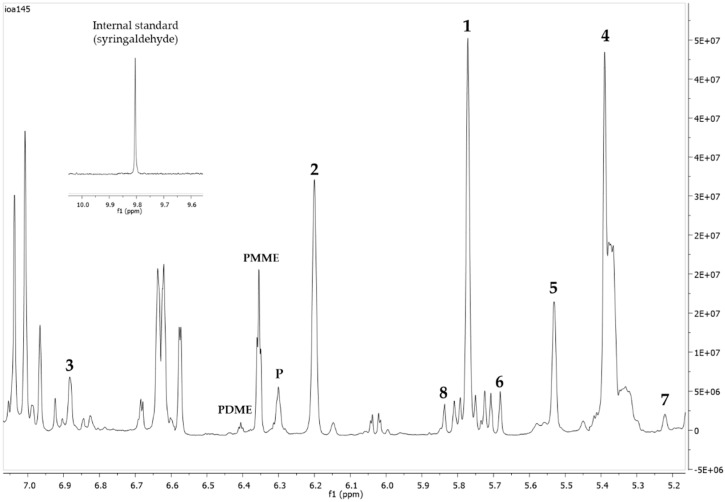
^1^H-NMR spectrum (400 MHz, CDCl_3_, δ-values in ppm) of black pine (P. nigra Arn.) heartwood extracts, showing the characteristic peaks of studied resin acids and internal standard. The numbers indicate the corresponding resin acids’ peaks used for quantitation: abietic acid (**1**), neoabietic acid (**2**), dehydroabietic acid (**3**), palustric acid (**4**), levopimaric acid (**5**), pimaric acid (**6**), sandaracopimaric acid (**7**), and isopimaric acid (**8**). Stilbenes’ peaks are also indicated (P = pinosylvin, PMME and PDME = monomethylether and dimethylether of pinosylvin, respectively).

**Figure 3 molecules-24-03603-f003:**
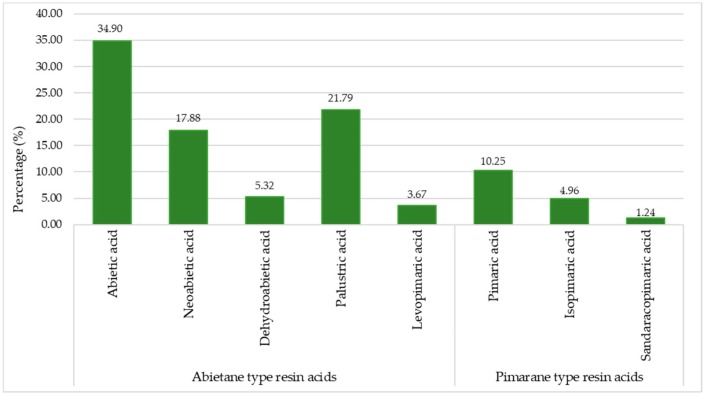
The mean percentage of abietane and pimarane types of resin acids.

**Figure 4 molecules-24-03603-f004:**
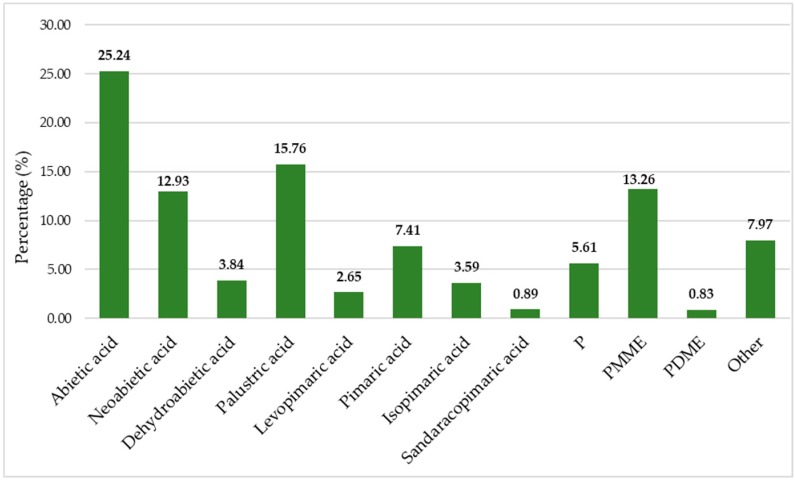
The mean percentage content of the constituents in the black pine’s heartwood acetone extraction from the Peloponnese (P = pinosylvin, PMME and PDME = monomethylether and dimethylether of pinosylvin, respectively).

**Figure 5 molecules-24-03603-f005:**
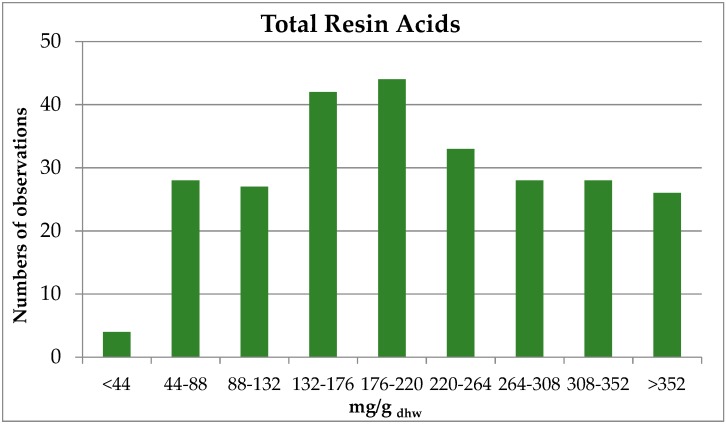
Total resin acids frequency distribution of black pine (*P. nigra* Arn.).

**Figure 6 molecules-24-03603-f006:**
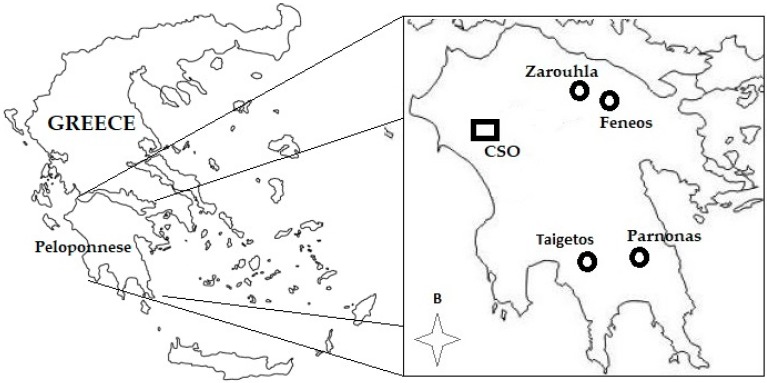
The positions of the provenances (circles) and the clonal seed orchard (CSO).

**Table 1 molecules-24-03603-t001:** ^1^H-NMR proton’s chemical shifts of the studied resin acids selected for quantitation (400 MHz, CDCl_3_, δ-values in ppm).

Resin Acid	Abietane Type					Pimarane Type		
Proton	Abietic Acid (1)	Neoabietic Acid (2)	Dehydroabietic Acid (3)	Palustric Acid (4)	Levopimaric Acid (5)	Pimaric Acid (6)	Sandaracopimaric Acid (7)	Isopimaric Acid (8)
**C(H)-15**	-	-	-	-	-	5.67	-	5.84
**C(H)-14**	5.77	6.20	6.88	5.39	5.53	-	5.22	-

**Table 2 molecules-24-03603-t002:** Recovery data and their coefficient of variance for the studied resin acids.

	Abietic Acid (1)	Neoabietic Acid (2)	Dehydroabietic Acid (3)	Palustric Acid (4)	Levopimaric Acid (5)	Pimaric Acid (6)	Sandaracopimaric Acid (7)	Isopimaric Acid (8)
**Recovery (%)**	90.26	93.52	88.78	90.50	90.73	92.75	92.07	87.37
**Coefficient of Variance (%)**	1.06	1.82	0.85	2.39	8.40	1.93	3.82	1.48

**Table 3 molecules-24-03603-t003:** Intraday precision data for the studied resin acids, expressed as relative standard deviations (%).

ID	Abietic Acid (1)	Neoabietic Acid (2)	Dehydroabietic Acid (3)	Palustric Acid (4)	Levopimaric Acid (5)	Pimaric Acid (6)	Sandaracopimaric Acid (7)	Isopimaric Acid (8)
Sample 1	5.55	6.29	8.84	2.17	9.07	5.09	9.82	9.07
Sample 2	6.51	7.38	3.49	5.18	4.85	7.55	2.88	3.31
Sample 3	3.68	5.40	2.48	9.92	8.81	7.66	2.98	7.57
Average	5.25	6.36	4.93	5.76	7.58	6.77	5.22	6.65

**Table 4 molecules-24-03603-t004:** The mean resin acid concentrations of *Pinus nigra* L. heartwood samples (*n* = 260) as determined by quantitative ^1^H-NMR.

	Abietic Acid (1)	Neoabietic Acid (2)	Dehydroabietic Acid (3)	Palustric Acid (4)	Levopimaric Acid (5)	Pimaric Acid (6)	Sandaracopimaric Acid (7)	Isopimaric Acid (8)	Total Resin Acids
Average mg/g_dhw_	76.77	39.34	11.69	47.94	8.07	22.54	2.72	10.91	219.98
Min. mg/g_dhw_	7.00	2.91	2.56	9.76	0.08	2.20	0.16	0.50	30.05
Max. mg/g_dhw_	181.75	101.82	38.59	105.22	64.91	59.42	6.67	34.09	424.70

**Table 5 molecules-24-03603-t005:** The mean resin acid concentrations (mg/g_dhw_) of the four *Pinus nigra* L. provenances’ heartwood after applying ANOVA and Duncan’s MRT at *p* = 0.05, Zarouhla: 65 samples, Feneos: 85 samples, Parnonas: 45 samples, and Taigetos: 65 samples, total 260 samples.

Provenance	Abietic Acid (1)	Neoabietic Acid (2)	Dehydroabietic Acid (3)	Palustric Acid (4)	Levopimaric Acid (5)	Pimaric Acid (6)	Sandaracopimaric Acid (7)	Isopimaric Acid (8)	Total Resin Acids
Zarouhla	81.99 ^a^	40.06 ^a^	11.10 ^a^	48.83 ^a^	7.41 ^a^	21.60 ^ab^	2.61 ^a^	11.45 ^a^	224.46 ^a^
Feneos	78.64 ^a^	37.84 ^a^	12.29 ^a^	46.43 ^a^	8.91 ^a^	25.09 ^a^	2.74 ^a^	11.03 ^a^	222.768 ^a^
Parnonas	79.99 ^ab^	41.87 ^a^	11.74 ^a^	49.86 ^a^	6.88 ^a^	23.57 ^ab^	2.92 ^a^	11.20 ^a^	227.83 ^a^
Taigetos	66.05 ^b^	36.36 ^a^	11.82 ^a^	45.61 ^a^	9.69 ^a^	20.35 ^b^	2.54 ^a^	9.89 ^a^	201.25 ^a^

The means followed by the same letter (a, b as superscript) are not statistically different.

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
