# Peer review of "High-Throughput 1H-Nuclear Magnetic Resonance-Based Screening for the Identification and Quantification of Heartwood Diterpenic Acids in Four Black Pine (Pinus nigra Arn.) Marginal Provenances in Greece"

_molecules, 2019, doi:10.3390/molecules24193603_

Round 1
Reviewer 1 Report
Sirs
The manuscript titled “High-Throughput 1H-Nuclear Magnetic Resonance based screening for the identification and quantification of heartwood diterpenic acids in four Black pine (Pinus nigra Arn.) marginal provenances in Greece” describes the composition of the acetone extract from Pinus Nigra heartwoods. Quantification is performed by 1H- NMR experiments and data are discussed in order to sort out the best chance for the production of bioactive substances.
The overall work would add information to an intriguing scientific field, however several points listed below need to be fixed before a possible recommendation for publication.
The introduction is appropriate but the last period is not clear: “Skakovskii et al. [36] stated that 13C-NMR spectroscopy was assumed to be more promising for analysis of such mixtures rather than 1H-NMR spectroscopy, but the use of NMR spectrometers with higher working frequencies make it possible to analyze resin acids using the 1H nuclei. Consequently, the identification of the analytes need less effort and time and is more convenient for quantitation [31,36].” The used spectrometer is a 400MHz whereas the Skakovskii studies exploit a 600 MHz so the “higher frequency” question is confusing and not acceptable. The 1H –NMR can be faster or more sensitive respect to 13C but it depends on so many factors (concentration, number of scans, overlaps, used relaxing agents….). We suggest a rephrasing with a new reference: “…..Skakovskii et al. [36], through the use of high field NMR spectrometers introduced the alternative 1H-NMR analysis of resin acids, this last technique is here preferred as quicker and straightforward. Moreover the possible simultaneous record of 1H-NMR and 13C-NMR experiments can pave the way to the robust analysis of resin acid by combined data as performed for other systems [Food Analytical Methods - https://doi.org/10.1007/s12161-019-01460-4].” Results: it is not clear whether numbers in the first paragraph are an average or not and also for figure 3 caption. Table 1 and Table 2 along with the numbers cited in pages 4 and 5 describe average values among several samples (I think 260, 65 and 20 often) and their inter-sample variability; it is not an “error” but an inter-group deviation (Table 2). Nonetheless all the values with uncertainty and variability should respect the significant figures and for sure there are too much uncertain digits: 76.77±37.35 is a nonsense, should be 80±40 or at least 77±37, 81.99(4.84) is 82(5). The table 2 caption about the “not statistically different” values is not clear, does it mean these values have a similar inter-provenance relative standard variability%? Values are all in mg/g out of the dry hartwood weight, we suggest to avoid putting always the acronym dhw or to use it always as subscript. Extraction protocol misses the acetone volume used for 200mg freeze dried mater. It should be constant in order to have the same extraction/extraction recovery By the way, it would be useful to know the real experimental error over measurements referred on the same sample. Actually should be 260 times 8 entries and maybe these could be in a supplementary material, also the recovery is supposed to have a kind of error, at least explain. Plant material: the acronym “ha” is not specified, to improve readability should be better years old or please explain first Please follow corrections suggested by the other reviewers; an overall English revision looks necessary, for instance the lines 257-262 should be rephrased
Author Response
Reviewer: The manuscript titled “High-Throughput 1H-Nuclear Magnetic Resonance based screening for the identification and quantification of heartwood diterpenic acids in four Black pine (Pinus nigra Arn.) marginal provenances in Greece” describes the composition of the acetone extract from Pinus Nigra heartwoods. Quantification is performed by 1H- NMR experiments and data are discussed in order to sort out the best chance for the production of bioactive substances.
The overall work would add information to an intriguing scientific field, however several points listed below need to be fixed before a possible recommendation for publication.
The introduction is appropriate but the last period is not clear: “Skakovskii et al. [36] stated that 13C-NMR spectroscopy was assumed to be more promising for analysis of such mixtures rather than 1H-NMR spectroscopy, but the use of NMR spectrometers with higher working frequencies make it possible to analyze resin acids using the 1H nuclei. Consequently, the identification of the analytes need less effort and time and is more convenient for quantitation [31,36].” The used spectrometer is a 400MHz whereas the Skakovskii studies exploit a 600 MHz so the “higher frequency” question is confusing and not acceptable. The 1H –NMR can be faster or more sensitive respect to 13C but it depends on so many factors (concentration, number of scans, overlaps, used relaxing agents….). We suggest a rephrasing with a new reference: “…..Skakovskii et al. [36], through the use of high field NMR spectrometers introduced the alternative 1H-NMR analysis of resin acids, this last technique is here preferred as quicker and straightforward. Moreover, the possible simultaneous record of 1H-NMR and 13C-NMR experiments can pave the way to the robust analysis of resin acid by combined data as performed for other systems [Food Analytical Methods - https://doi.org/10.1007/s12161-019-01460-4].”
Reviewer: Results: it is not clear whether numbers in the first paragraph are an average or not and also for figure 3 caption. Table 1 and Table 2 along with the numbers cited in pages 4 and 5 describe average values among several samples (I think 260, 65 and 20 often) and their inter-sample variability; it is not an “error” but an inter-group deviation (Table 2).
Authors reply: In tables 4 and 5 we have clarified that the given results concern the mean concentration of the 260 samples of the studied resin acids. To avoid confusion and according to reviewer 4 the SD values have been removed and we have kept only the min-max values. Indeed the previously presented values corresponded to the variation among the 260 samples.
Reviewer: Nonetheless all the values with uncertainty and variability should respect the significant figures and for sure there are too much uncertain digits: 76.77±37.35 is a nonsense, should be 80±40 or at least 77±37, 81.99(4.84) is 82(5).
Authors reply: According to the equation for quantitative determination of the resin acids in paragraph “4.4 1H-NMR spectral analysis/quantitation methodology”, the numbers of significant figures are defined by IRA, msample and RRA. Every parameter in the equation was measured using four significant figures (i.e. weight msample =200.0 mg, IRA = 23.45, RRA =90.26%). Although it seems that we have used a big number of uncertain digits, in fact we have taken care to use four significant figures in every step of measurements.
Reviewer: The table 2 caption about the “not statistically different” values is not clear, does it mean these values have a similar inter-provenance relative standard variability%?
Authors reply: The inter provenance variance have similar values as one of the ANOVA’s assumptions is the homogeneity of variances among provenances’ samples (homoscedasticity). We have used Duncan’s Multiple Range Test, a post hoc test, to detect if there were differences among the provenance concentrations’ means.
Reviewer: Values are all in mg/g out of the dry hartwood weight, we suggest to avoid putting always the acronym dhw or to use it always as subscript.
Authors reply: We have now used always as subscript.
Reviewer: Extraction protocol misses the acetone volume used for 200mg freeze dried mater. It should be constant in order to have the same extraction/extraction recovery
Authors reply: The acetone volume used in the extraction process (12 mL) was added in paragraph “4.3 Extraction protocol”, and was constant for every sample of dried heartwood in order to achieve the same recovery.
Reviewer: By the way, it would be useful to know the real experimental error over measurements referred on the same sample. Actually should be 260 times 8 entries and maybe these could be in a supplementary material, also the recovery is supposed to have a kind of error, at least explain.
Authors reply: We have calculated the experimental error in three random samples analyzed three times each in the same day, presented now in Table 3. The complete results for all compounds in the three random samples are presented in the supplementary material (Table 1) confirming the precision of the method. In all cases the experimental error was less than 10% (Table 3). Due to high precision of the method, we did not perform triplicate analysis for the whole number of samples. Moreover, we have added the coefficient of variance for the recovery data (Table 2) in our manuscript, as well as additional data presented in supplementary material (Table 2).
Reviewer: Plant material: the acronym “ha” is not specified, to improve readability should be better years old or please explain first
Authors reply: ha = hectare at paragraph “4.1 Plant material”
Reviewer: Please follow corrections suggested by the other reviewers; an overall English revision looks necessary, for instance the lines 257-262 should be rephrased.
Authors reply: We rephrased the lines 257-262 (now 276-281) and we have also corrected several points throughout the manuscript concerning the English language.
Reviewer 2 Report
This manuscript offers some interesting results on the composition in diterpenic resin acids of Black pine heartwood. The applied technique was the 1H-NMR spectroscopy that highlights its feasibility and wide practical applicability in characterizing various complex matrixes, from leaves [1] and walnuts [2] for profiling “green” extracts and different genotypes, respectively, to foodstuff [3, 4] for quality/safety or discrimination purposes, and biological [5] for lipid profiling.
Antonietta Cerulli, Milena Masullo, Paola Montoro, Jan Hošek, Cosimo Pizza, Sonia Piacente, Metabolite profiling of “green” extracts of Corylus avellana leaves by 1H NMR spectroscopy and multivariate statistical analysis, Journal of Pharmaceutical and Biomedical Analysis, Volume 160, 25 October 2018, Pages 168-178 Raluca Popescu, Roxana Elena Ionete, Oana Romina Botoran, Diana Costinel, Felicia Bucura, Elisabeta Irina Geana, Yazan Falah Jadee ’Alabedallat and Mihai Botu, 1H-NMR Profiling and Carbon Isotope Discrimination as Tools for the Comparative Assessment of Walnut (Juglans regia L.) Cultivars with Various Geographical and Genetic Origins—A Preliminary Study, Molecules 2019, 24, 1378; doi:10.3390/molecules24071378 Donny W. H. Merkx, G. T. Sophie Hong, Alessia Ermacora, John P. M. van Duynhoven, Rapid Quantitative Profiling of Lipid Oxidation Products in a Food Emulsion by 1H NMR, Anal. Chem.2018, 90, 7, 4863-4870, https://doi.org/10.1021/acs.analchem.8b00380 Shuangxi Fan, Qiding Zhong, Carsten Fauhl-Hassek, Michael K.-H. Pfister, Bettina Horn, Zhanbin Huang, Classification of Chinese wine varieties using 1H NMR spectroscopy combined with multivariate statistical analysis, Food Control, 2018, Volume 88, Pages 113-122, https://doi.org/10.1016/j.foodcont.2017.11.002 Rubén Barrilero, Miriam Gil, Núria Amigó, Cintia B. Dias, Lisa G. Wood, Manohar L. Garg, Josep Ribalta, Mercedes Heras, Maria Vinaixa, Xavier Correig, LipSpin: A New Bioinformatics Tool for Quantitative 1H NMR Lipid Profiling, Anal. Chem.2018, 90, 3, 2031-204, https://doi.org/10.1021/acs.analchem.7b04148
For the study, a number of 260 samples of Pinus nigra clones, originated from Peloponnese, Greece, were selected for the study. Consequently, the environmental effect could be studied.
Some point comments:
Introduction
In the introduction, please insert few words related to the 1H-NMR technique. Also, try to find also more recent data/references since the reported bibliography is relative not new.
Figure 2 should be placed to subchapter 4.2. Sampling.
Results
Figure 3 can be removed since no new information is revealed compared with the text “The resin acid fraction accounted for 68.29% of the totalextractive content, while stilbenes were presented in significant quantities comprising 19.53%. The rest 12.18% is referred to other substances presented in TAE, i.e. fatty acids, unsaponifiables, triglycerides, other phenols, waxes, sterols etc.”
The same for Figures 4 – 6. They are redundant with the information in text. You can reconsider to explain only in the text or to make a synthetic table with the data represented in the Figures 4 – 6.
Results: Data presented are means; some individual data would be interesting, even if presented as supplementary material.
Conclusions are to general, thus can be improved.
Author Response
Reviewer: This manuscript offers some interesting results on the composition in diterpenic resin acids of Black pine heartwood.
The applied technique was the 1H-NMR spectroscopy that highlights its feasibility and wide practical applicability in characterizing various complex matrixes, from leaves [1] and walnuts [2] for profiling “green” extracts and different genotypes, respectively, to foodstuff [3, 4] for quality/safety or discrimination purposes, and biological [5] for lipid profiling.
Antonietta Cerulli, Milena Masullo, Paola Montoro, Jan Hošek, Cosimo Pizza, Sonia Piacente, Metabolite profiling of “green” extracts of Corylus avellana leaves by 1H NMR spectroscopy and multivariate statistical analysis, Journal of Pharmaceutical and Biomedical Analysis, Volume 160, 25 October 2018, Pages 168-178
Raluca Popescu, Roxana Elena Ionete, Oana Romina Botoran, Diana Costinel, Felicia Bucura, Elisabeta Irina Geana, Yazan Falah Jadee ’Alabedallat and Mihai Botu, 1H-NMR Profiling and Carbon Isotope Discrimination as Tools for the Comparative Assessment of Walnut (Juglans regia L.) Cultivars with Various Geographical and Genetic Origins—A Preliminary Study, Molecules 2019, 24, 1378; doi:10.3390/molecules24071378
Donny W. H. Merkx, G. T. Sophie Hong, Alessia Ermacora, John P. M. van Duynhoven, Rapid Quantitative Profiling of Lipid Oxidation Products in a Food Emulsion by 1H NMR, Anal. Chem.2018, 90, 7, 4863-4870, https://doi.org/10.1021/acs.analchem.8b00380
Shuangxi Fan, Qiding Zhong, Carsten Fauhl-Hassek, Michael K.-H. Pfister, Bettina Horn, Zhanbin Huang, Classification of Chinese wine varieties using 1H NMR spectroscopy combined with multivariate statistical analysis, Food Control, 2018, Volume 88, Pages 113-122, https://doi.org/10.1016/j.foodcont.2017.11.002
Rubén Barrilero, Miriam Gil, Núria Amigó, Cintia B. Dias, Lisa G. Wood, Manohar L. Garg, Josep Ribalta, Mercedes Heras, Maria Vinaixa, Xavier Correig, LipSpin: A New Bioinformatics Tool for Quantitative 1H NMR Lipid Profiling, Anal. Chem.2018, 90, 3, 2031-204, https://doi.org/10.1021/acs.analchem.7b04148
Authors reply: According to the reviewer’s suggestion we have included in our manuscript the proposed recent references
Reviewer: For the study, a number of 260 samples of Pinus nigra clones, originated from Peloponnese, Greece, were selected for the study. Consequently, the environmental effect could be studied.
Authors reply: We thank the reviewer for the remarkable comment and we want to report that the environmental effect as some genetic parameters e.g. heritability, will presented in subsequent paper we are working on.
Reviewer:
Some point comments:
In the introduction, please insert few words related to the 1H-NMR technique. Also, try to find also more recent data/references since the reported bibliography is relative not new.
Authors reply: According to the reviewer’s suggestion we have included in our manuscript the proposed recent references and a sort text about recent applications of the NMR technique (line 95-100).
Reviewer: Figure 2 should be placed to subchapter 4.2. Sampling.
Authors reply: Figure 2 has been now placed to subchapter 4.2 as Figure 6
Reviewer:
Figure 3 can be removed since no new information is revealed compared with the text “The resin acid fraction accounted for 68.29% of the total extractive content, while stilbenes were presented in significant quantities comprising 19.53%. The rest 12.18% is referred to other substances presented in TAE, i.e. fatty acids, unsaponifiables, triglycerides, other phenols, waxes, sterols etc.”
The same for Figures 4 – 6. They are redundant with the information in text. You can reconsider to explain only in the text or to make a synthetic table with the data represented in the Figures 4 – 6.
Authors reply: Figures 3, 4 and 6 have been moved to the supplementary material. Figure 5 has been changed to figure 3 and Figure 7 to figure 4. For both figures we modified the chart type. We kept some Figures for a more comprehensible visual representation of the text, as proposed by reviewer 4.
Reviewer: Data presented are means; some individual data would be interesting, even if presented as supplementary material.
Authors reply: According to others reviewers we have added some data about individual samples that are presented in the supplementary material (Tables 1 and 2).
Reviewer: Conclusions are to general, thus can be improved.
Authors reply: We have rephrased and improved the conclusion.
Reviewer 3 Report
The manuscript from Ioannidis et al. describes the application of 1H NMR spectroscopy for the identification and quantification of heartwood diterpenic acids in black pine.
The study is interesting and the manuscript is well written, therefore I recommend it for publication in Molecules.
I only have a couple of minor comments/questions:
1) Figure 10 shows a spectrum in the range of 7.1-5.2 ppm. This is fine since the signals of interest appear in that area. However, a whole spectrum needs to be shown not necessarily in the main text but at least as supporting information.
2) A 10 s delay for magnetization recovery seems reasonable, but is this just a guess or authors performed any T1 measures?
Author Response
Reviewer: The manuscript from Ioannidis et al. describes the application of 1H NMR spectroscopy for the identification and quantification of heartwood diterpenic acids in black pine.
The study is interesting and the manuscript is well written, therefore I recommend it for publication in Molecules.
I only have a couple of minor comments/questions:
1) Figure 10 shows a spectrum in the range of 7.1-5.2 ppm. This is fine since the signals of interest appear in that area. However, a whole spectrum needs to be shown not necessarily in the main text but at least as supporting information.
Authors reply: A whole spectrum is shown in the supplementary material Figure1 as supporting information.
Reviewer: 2) A 10 s delay for magnetization recovery seems reasonable, but is this just a guess or authors performed any T1 measures?
Authors reply: We performed a series of experiments, trying several delays, ranging from one to ten seconds, in steps of 1 second, until we achieved constant integration values for all target compounds in comparison.
Reviewer 4 Report
The manuscript “High-Throughput 1H-Nuclear Magnetic Resonance-based screening for the identification and quantification of heartwood diterpenic acids in four Black pine (Pinus nigra Arn.) marginal provenances in Greece” was submitted to Molecules for publication. The study describes analysis of 260 pine samples by quantitative NMR and the geographic variation of their contents.
Broad comments:
First of all, I must say that the topic of the study is very interesting, and the number of samples measured, in general, allows profound conclusions. However, two major comments must be addressed. The first point is the organization and design of the manuscript (which I will refer to later), which needs improvement, and the second harder to accomplish point is the validity of the used NMR method. Analytical method development and validation is the basis of good research and thus should be paid primary attention to.
In the case of the present work, the authors claim that “nonoverlapping, undoubtedly defined peaks” were selected for quantification. However, a look at the spectrum makes clear that this is definitely not the case. Peak 3 has a strange peak shape, peak 4 is overlapping, peak 5 should actually be a singulett due to the lack of neighbouring protons both rather looks like a broad multiplett, what might be the reason for the very low recovery rates of levopimaric acid. Therefore, a two-dimensional spectrum would be of help, though there might still be an overlap of signals. Another point is that the authors did not conduct precision measurements, which would have been at least the second core parameter to assess the validity of the method.
Regarding the design of the manuscript several changes have to be made. Figure 1:
Please use the general numbering for diterpenes as shown in e.g. Liebigs Ann. Chem. 1992, 911 -919. Please give compound numbers in brackets and display the compounds according to the compound numbers.
Figures 3 to 6: The “cake diagrams” look kind of strange in a chemical journal and should be substituted with standard diagrams. Also think about reducing them, e.g. Figure 3 and 6 don’t really give a lot of information.
Figure 8 is not necessary as 8a to 8h all have more or less the same pattern. Thus, showing only figure 9 is sufficient.
Figure 10: This figure, as well as description / validation of the method should move to the beginning of the results section. Furthermore, also the whole spectrum should be displayed, e.g. divide into Figure 10a (whole spectrum) and Figure 10b (zoom from 5 to 7 ppm). Also indicate field strength and solvent used in the figure legend. Also, show a 2D spectrum in order to exclude peak overlap not visible in the 1D spectrum (as explained above) and chose other signals for at least compound 4 and 5.
Table 1: What exactly should this table tell the reader? If it is precision the method is not valid (RSD values of more than 50%). If it is just the variation among the 260 species (what I think), please don’t use SD as it is confusing. Rather only show min and max values or a range.
Introduction: The introduction is very informative but a little bit too long and could tire the reader. Therefore, the biosynthesis part should be drastically shortened (as it is basic knowledge). I would also suggest moving the last part of the introduction (line 81 to 86) forward, as it does not fit where it stands now.
Minor comments:
Please also always write the word acid and not only e.g. pimaric or abietic.
Also consult a native speaker as some text passages are difficult to read.
Author Response
Reviewer 4
Reviewer: The manuscript “High-Throughput 1H-Nuclear Magnetic Resonance-based screening for the identification and quantification of heartwood diterpenic acids in four Black pine (Pinus nigra Arn.) marginal provenances in Greece” was submitted to Molecules for publication. The study describes analysis of 260 pine samples by quantitative NMR and the geographic variation of their contents.
Broad comments:
First of all, I must say that the topic of the study is very interesting, and the number of samples measured, in general, allows profound conclusions. However, two major comments must be addressed. The first point is the organization and design of the manuscript (which I will refer to later), which needs improvement, and the second harder to accomplish point is the validity of the used NMR method. Analytical method development and validation is the basis of good research and thus should be paid primary attention to.
In the case of the present work, the authors claim that “nonoverlapping, undoubtedly defined peaks” were selected for quantification. However, a look at the spectrum makes clear that this is definitely not the case. Peak 3 has a strange peak shape, peak 4 is overlapping, peak 5 should actually be a singulett due to the lack of neighbouring protons both rather looks like a broad multiplett, what might be the reason for the very low recovery rates of levopimaric acid. Therefore, a two-dimensional spectrum would be of help, though there might still be an overlap of signals.
Authors reply: We would like to thank reviewer 4 for the careful observation about the partial overlapping of peak 4. For the specific compound (palustric acid) there is no other peak without overlapping and for this reason we chose the peak with the less possible overlapping. Due to the fact that the peak at 5.39 ppm is a sharp singlet we used only the integration value of the non overlapping part and we compared the integration value with that obtained by the deconvolution spectra (see supplementary Figure 4). The obtained values of integration, were very similar with always less than 5% difference and for this reason we followed the above described methodology for integration.
As proposed by the reviewer we present in the supplementary file a 2D HMQC spectrum (Figure 2) where it is clear that neither peak 3 nor peak 5 present any overlapping. Especially for peak 5 (levopimaric acid) we present a new 1H-NMR spectrum where it clearly presents a sharp singlet peak (Figure 3 in the manuscript). The previous figure with H-NMR spectrum was from a sample with very low concentration of levopimaric acid and for this reason it was replaced by a new one. The 2D HMQC spectrum of the sample corresponding to the old figure is presented in supplementary material (Figure 3) as a second example.
We would like to note that the observation of the reviewer helped us to clarify a small number of samples where the peak of levopimaric acid and isopimaric acid had not been integrated correctly and this has led to a change in the presented values of concentration, without affecting the overall results. We are really grateful for this observation since it helped us to ameliorate the quality of the manuscript.
Reviewer: Another point is that the authors did not conduct precision measurements, which would have been at least the second core parameter to assess the validity of the method.
Authors reply: The precision measurements have been added and are now presented in Table 3 and supplementary table 1.
Reviewer: Figure 1: Please use the general numbering for diterpenes as shown in e.g. Liebigs Ann. Chem. 1992, 911 -919. Please give compound numbers in brackets and display the compounds according to the compound numbers.
Authors reply: We have changed the C numbering according to proposed literature, i.e. Liebigs Ann. Chem. 1992, 911 -919, giving the numbers of each studied resin acid in brackets (Figure 1).
Reviewer: Figures 3 to 6: The “cake diagrams” look kind of strange in a chemical journal and should be substituted with standard diagrams. Also think about reducing them, e.g. Figure 3 and 6 don’t really give a lot of information.
Authors reply: Figures 3, 4 and 6 have been moved to the supplementary material. Figure 5 has been changed to figure 3 and Figure 7 to figure 4. For both figures we modified the chart type.
Reviewer: Figure 8 is not necessary as 8a to 8h all have more or less the same pattern. Thus, showing only figure 9 is sufficient.
Authors reply: Figures 8a to 8h were transferred to supplementary material. We have kept only figure 9 (now figure 5) as proposed.
Reviewer: Figure 10: This figure, as well as description / validation of the method should move to the beginning of the results section.
Authors reply: We have moved figure 10 to the proposed section with a new number figure 2.
Reviewer: Furthermore, also the whole spectrum should be displayed, e.g. divide into Figure 10a (whole spectrum) and Figure 10b (zoom from 5 to 7 ppm).
Authors reply: As proposed by reviewer 3, we added a whole spectrum in supplementary material (Figure 1)
Reviewer: Also indicate field strength and solvent used in the figure legend.
Authors reply: We have added the strength and the used solvent in all NMR spectra Figures.
Reviewer: Also, show a 2D spectrum in order to exclude peak overlap not visible in the 1D spectrum (as explained above) and chose other signals for at least compound 4 and 5.
Authors reply: We added two 2D HMQC spectra of two different samples as supplementary material (Figures 2 and 3) to demonstrate that the selected peaks present no or very small overlapping not significantly affecting the obtained results.
Reviewer: Table 1: What exactly should this table tell the reader? If it is precision the method is not valid (RSD values of more than 50%). If it is just the variation among the 260 species (what I think), please don’t use SD as it is confusing. Rather only show min and max values or a range.
Authors reply: In Table 1, it is not indeed the RSD values but the variation among the 260 samples (standard deviation, i.e. s, of the samples). We have erased the SD values, keeping the min and max values.
Reviewer: Introduction: The introduction is very informative but a little bit too long and could tire the reader. Therefore, the biosynthesis part should be drastically shortened (as it is basic knowledge). I would also suggest moving the last part of the introduction (line 81 to 86) forward, as it does not fit where it stands now.
Authors reply: Overall, as proposed by others reviewers too, we have rearranged and shortened several parts. The proposed paragraph has been moved at the beginning of the introduction.
Reviewer: Please also always write the word acid and not only e.g. pimaric or abietic.
Authors reply: We have added the word acid after each resin acid.
Reviewer: Also consult a native speaker as some text passages are difficult to read.
Authors reply: we have corrected several points throughout the manuscript concerning the English language.
Round 2
Reviewer 1 Report
Sirs, looks like all the comments have been held into consideration and the paper can be recommended fo publication from this side
Reviewer 4 Report
The authors adhered to all the suggestions made resulting in a manuscript of much higher quality and readability.
The only error still to be corrected is the numbering of the compounds in Figure 1. Numbering of C1 and C4 are confused and the methyl group attached to C10 must be numbered C20.